# Effects of Rare Earth Elements on Blood Pressure and Their Exposure Biomarkers: Evidence from Animal Experiments

**DOI:** 10.3390/ijerph18189836

**Published:** 2021-09-18

**Authors:** Yiming Pang, Jianjun Jiang, Kexin Li, Lailai Yan, Yanqiu Feng, Junli Wang, Xiaolu Cao, Zhiwen Li, Bin Wang

**Affiliations:** 1Key Laboratory of Reproductive Health, National Health Commission of the People’s Republic of China, Institute of Reproductive and Child Health, Peking University, Beijing 100191, China; 1911110148@pku.edu.cn (Y.P.); 1610306103@pku.edu.cn (Y.F.); 2Department of Epidemiology and Biostatistics, School of Public Health, Peking University, Beijing 100191, China; 3Department of Toxicology, School of Public Health, Peking University, Beijing 100191, China; jiangjj_bmu@pku.edu.cn; 4Institute of Geographic Sciences and Natural Resources Research, CAS, Beijing 100101, China; likx@igsnrr.ac.cn; 5Department of Laboratorial Science and Technology, School of Public Health, Peking University, Beijing 100191, China; yll@bjmu.edu.cn; 6Basic Clinical Laboratory Teaching and Research Department, School of Medical Laboratory, Youjiang Medical University for Nationalities, Baise 533000, China; baisewangjunli@ymcn.edu.cn; 7Hubei Province Key Laboratory of Occupational Hazard Identification and Control, Department of Environmental Health and Occupational Medicine, School of Public Health, Wuhan University of Science and Technology, Wuhan 430081, China; caoxiaolu@wust.edu.cn

**Keywords:** rare earth elements, hair, blood pressure, biomarker, oxidative stress

## Abstract

Solid fuel combustion is an important source of the release of rare earth elements (*REEs*) into the ambient environment, resulting in potential adverse effects on human cardiovascular health. Our study aimed to identify reliable exposure biomarkers of REE intake and their potential role in blood pressure change. A total of 24 rats were administered with 14 REE chlorides at four doses (six rats per group). Fur samples were collected both before and after administration. Blood samples were collected after 12 weeks of REE intake. The REE concentrations in rat fur and blood samples were measured by inductively coupled plasma mass spectrometry. For each week, blood pressure, as well as heart rate and pulse pressure, were measured. The linear mixed-effect model was used to analyze the relationship between REE administration dose and blood pressure change. We found that the REE concentration in fur, but not blood, samples exhibited significant dose–response relationships with administration dose. It suggested that hair samples are a more efficient matrix for indicating the exposure level of a population to *REEs* than blood samples. However, there was no dose–response relationships between the administration dose and blood pressure change of rats, or with heart rate and pulse pressure for the 14 *REEs*. We also did not find a dose–response relationship between REE administration levels and plasma concentration of 8-hydroxy-2’-deoxyguanosine, as an important DNA oxidative stress damage biomarker. In conclusion, hair samples are more suitable as a sample type to reliably assess exposure to *REEs* than blood samples, and *REEs* did not have a direct adverse effect on blood pressure in our rat model.

## 1. Introduction

Rare earth elements (*REEs*) include 17 elements in the lanthanide series, which have similar physicochemical properties. *REEs* have become essential components in modern human life due to their increasing number of applications in technology, for example, in cell phones, laptops, cruise missiles, radar systems, and reactive armor [1]. China has the world’s largest reserve of REE resources and dominates international market supplies [2]. *REEs* can enter the environment through several routes. For example, some *REEs* are used as important trace fertilizers in order to stimulate crop growth in China [3,4]. Additionally, coal mining activities may contaminate ambient soil or water with *REEs* [5]. Furthermore, coal combustion fly ash contains high amounts of *REEs*, and therefore, fly ash is a common source of air pollution in northern China [6,7]. It has been reported that there is positive association between indoor air pollution from coal combustion and concentrations of *REEs* in hair among women living in Shanxi Province, China [8]. In addition, high intake of *REEs* might increase the risk of hypertension among housewives [9]. In China, overall, 23.2% (~244.5 million) of the Chinese adult population ≥18 years of age suffer from hypertension, while another 41.3% (~435.3 million) suffer from pre-hypertension, as defined by the Chines guidelines [10]. Physiological changes in blood pressure, as well as hypertension, have been investigated for decades, and environmental factors are considered to play an important role in these conditions. Among them, the toxic effects of *REEs* on public health are regarded as a public concern [9,11].

The biological effects of *REEs* have been widely reported. It seems that *REEs* have a “hormetic effect” with respect to their dose–response phenomenon, producing health benefits at low concentrations while causing damage at high intake. High REE exposure can cause elevated inflammatory effect [12], immune function damage [13], oxidative stress damage [14], and reproductive toxicity [11]. Research using a zebrafish model has indicated the adverse effects of *REEs* on swimming behavior and cardiac morphology, especially in the case of La and Pr, where cardiac hypertrophy and cardiac muscle contraction were two important toxicity pathways [15]. In contrast, protective effects of *REEs* have also been reported. For example, the rat mode indicated that lanthanum nitrate can ameliorate atherosclerosis by regulating lipid metabolism, oxidative stress, endothelial dysfunction and inflammatory response in mice, and its potential mechanism is associated with the inhibition of MAPK and NF-κB signaling pathways [16]. In addition, lanthanum nitrate could restore the redox homeostasis disrupted by ethanol by provoking the Keap 1/Nrf2/p62 signaling pathway [17]. It has been recognized that chronic inflammation and oxidative stress both play key roles in the mediation of damage to the arterial wall [18], further increasing the risk of hypertension [19]. In vitro experiments have indirectly proved the possible association between *REEs* and cardiovascular disease, but this is supported by few epidemiological studies, there is no evidence from animal models. In addition, REE concentrations in hair and blood samples have both been described as reliable sample types for representing REE intake in the general population [9,11,20,21]. However, there is a lack of animal experimental data to address the rationality of choosing this biological matrix. Moreover, hair samples may be contaminated by ambient particulate matter, resulting in a certain difficulty in building a relationship between REE intake in the general population and their REE concentration in hair. In our study, using a rat model, we aimed to: (1) screen the reliable exposure biomarkers of *REEs*, and (2) examine the effects of REE intake on blood pressure.

## 2. Materials and Method

### 2.1. Reagents

A total of 14 REE chlorides were bought from InnoChem Co., Beijing, China, including lanthanum (La), cerium (Ce), praseodymium (Pr), neodymium (Nd), samarium (Sm), europium (Eu), gadolinium (Gd), terbium (Tb), dysprosium (Dy), holmium (Ho), erbium (Er), thulium (Tm), ytterbium (Yb), and yttrium (Y). Highly sensitive 8-OHdG check (KOG-HS10E) was purchased from Japan Institute for the Control of Aging, Japan. Nitric acid was obtained from Anpel Laboratory Technologies, Shanghai, China. Female rats (SPF grade, Sprague-Dawley) were obtained from the Department of Laboratory Animal Science, Peking University Health Science Center.

### 2.2. Rat Model

Twenty-four female rats with body weight range of 180–220 g were fed in sterile plastic cages at a biosafety laboratory (BSL-2) under room temperature (23 ± 3 °C), 40–60% humidity, and a 12-h light/dark cycle. ^60^Co-radiation-sterilization laboratory feedstuff (Huafukang, Inc., Beijing, China) and sterile water were provided ad libitum. The rats were randomly divided into four administration dose groups (6 rats per group) after a week of adaptive feeding. A total of 14 REE chlorides dissolved in ultrapure water were administered to the rats, including La, Ce, Pr, Nd, Sm, Eu, Gd, Tb, Dy, Ho, Er, Tm, Yb, and Y. Chlorinated lutetium (Lu) was unavailable and therefore not included in the study. Four administration doses were administrated using saline solution (Control), 200 times concentration of human estimated daily intake [22] of *REEs* (Low), 2000 times concentration (Middle), and 20,000 times concentration (High). In the low-, middle- and high-exposure groups, REE chlorides dissolved in ultrapure water were diluted to different concentration, and for the control group, only the saline solution was used for administration. A schematic of the study is provided in Figure 1. Before administration, the fur was cut off from the back as close to the skin as possible (Stage-I). Then, all rats were dosed with intragastric administration for 12 weeks. The administration doses at each time are provided in Appendix A. The setting of relative concentrations among various *REEs* was performed with reference to the previous study about the dietary intake and burden of lanthanide for a Chinese population [22]. For each rat, a 2-mL dosing solution was administered. Every week, the rats were dosed three times in the afternoon on Monday, Wednesday, and Friday, respectively. We benefitted from the advantages of non-invasive collection; fur samples from rats’ back were collected during week 1 (start of Stage-I), week 6 (end of Stage-II) and week 12 (end of Stage-III), respectively, and stored at −80 °C prior to experimental analysis. For each week (excluding 3 weeks with equipment failure), we measured the blood pressure, as well as the heart rate, body weight, and pulse pressure. Limited by available measurement equipment and experiment condition, three fixed rats in each group were monitored using a noninvasive sphygmomanometer (Model: BP-2010A, Softron, China). After 12-week administration, the blood samples were collected from the heart for all rats and transferred into anticoagulant tubes to separate the plasma and blood cells by centrifugation at 3000 rpm for 15 min. All animal care and experimental procedures involved in the present study strictly followed the Guide for the Care and Use of Laboratory Animals. This study was approved by the Committee on the Ethics of Animal Experiments of the Peking University Health Science Center.

### 2.3. Quantitative Analysis

***REEs in rat blood.*** From each rat, 0.2 g blood cells and 0.2 mL plasma samples were transferred to microwave reaction tanks, and 400 μL 50% (*v*/*v*) nitric acid and 400 μL ultrapure water were added to each sample, followed by a digestion procedure using a microwave reaction system (MARS6 Model, CEM, Matthews, USA). REE concentrations in the digestion solution were measured by high-resolution ICP-MS (ELAN DRC-e, Perkin Elmer, Billerica, USA) with the following parameters: nebulizer gas flow, 0.84 L/min; auxiliary gas flow, 1.20 L/min; plasma gas flow, 15.0 L/min; radio frequency generator power, 1.10 KW; dwell time, 50 ms; mode peak, hopping; resolution, 0.6–0.8 amu.

***REEs in rat fur.*** The detailed procedure for fur washing and digestion has been described elsewhere [23]. Briefly, a washing protocol including sodium dodecyl sulfate, Triton, methanol, and a water bath was adopted. The fur samples were digested by nitric acid using the microwave reaction system (MARS6 Model, CEM). The REE concentration in the digestion solution was measured by high-resolution ICP-MS (ELAN DRC-e, Perkin Elmer). The parameters used have been described previously. The information on the quality control and detection limit of ICP-MS were provided in Appendix A.

***8-hydroxy-2’-deoxyguanosine (8-OHdG) in Plasma.*** 8-OHdG is a product of oxidatively damaged DNA formed by hydroxy radical, singlet oxygen, and direct photodynamic action. Rat’s plasma 8-OHdG concentration was measured by following the instruction of a highly sensitive 8-OHdG check. The measure theory is ELISA method based on the competitive inhibition immunochromatography mechanism. The regression coefficients of the standard curves were all above 0.99.

### 2.4. Data Analysis

The mean and standard deviation (SD) were used to describe the overall distribution of the concentrations of *REEs* and blood pressure. Linear regression was used to analyze the concentration trend of different exposure level. Spearman correlation analysis was used to describe the correlations between REE administration level and plasma 8-OHdG. According to the repeated measurement data of rat blood pressure during the administration period, a linear mixed-effect model was used to analyze the associations between REE administration level and blood pressure as below,
*Y* = *β* × *REEs*+ *k* × *BW* + *γ*(*S*)(1)
where *Y* is the dependent variable, e.g., rat blood pressure, pulse pressure, and heart rate, *REEs* is the REE administration level as the fixed term with the coefficients of *β*; *BW* represents rat body weight as the potential confounder with the coefficient of *k*; *γ(S)* is a random intercept for each rat. A value of *p* < 0.05 (two-sided) is considered to indicate statistical significance. The regression coefficient *β* and its 95% confidence intervals (CIs) represent the association between *REEs* administration dose and blood pressure in rats. All statistical analyses were performed using R software package *lme4* (v. 3.6.1; R Core Team, Auckland, New Zealand).

## 3. Results

### 3.1. Exposure Biomarkers of REEs in Rats

The average body weight changes of the four groups are shown in Figure 2, and the detailed information can be found in Appendix A. It suggests that REE administration did not significantly affect the body weight of rats in the REE administration groups.

For the fur sample collected at Stage-I, there were no statistical differences in the four administration groups for any of the *REEs*, except for Y (see Table 1). For the Y element, a relatively higher concentration was found in the control group (i.e., 1002 ± 486 ng/g) than in the other three groups—Low (423.5 ± 138.3 ng/g), Middle (480 ± 227 ng/g), and High (285 ± 205 ng/g)—and an even higher concentration was found compared to the control groups at Stage-II (549 ± 113 ng/g) and Stage-III (500 ± 697 ng/g). This could be due to outliers in the measurement. Overall, we consider that the 24 rats could be evenly divided into four groups. For the fur samples collected at both Stage-II and Stage-III, there were significant overall dose–response relationships for the 14 administered *REEs* in the fur samples (Table 1). In contrast, no dose–response relationships were observed between the doses of the 14 *REEs* administered to rats and their concentrations in either the plasma or blood samples (Table 2). This suggests that these two biological sample types are not reliable indicators of exposure.

### 3.2. Effects of REE Intake on Blood Pressure

Using linear mixed-effect regression analysis, we were not able to observe a significant dose–response relationship between the rats’ REE administration dose and blood pressure, i.e., systolic and diastolic blood pressures. Additionally, there were no associations between administration dose of *REEs* and the other physiological indicators of heart rate and pulse pressure (see Table 3).

### 3.3. Correlation between REE Administration and Oxidative Stress Damage

We further analyzed the 8-OHdG concentration in rat plasma to represent their oxidative stress damage level (see Figure 3). It seems that there was a slight increase of plasma 8-OHdH with increasing REE administration dose. However, the Spearman correlation analysis showed that there was no significant correlation between REE administration dose and 8-OHdG concentration in rat plasma (*r* = 0.303, *p* = 0.160).

## 4. Discussion

In this study, we firstly evaluated the effect of high doses of REE administration on blood pressure using a rat model, and found that REE exposure may not have a significant effect on changes in blood pressure. We also confirmed that hair samples may be more appropriate than blood samples for evaluating the exposure level to *REEs*.

To date, only a small number of studies in the literature have analyzed the dose–response relationships between REE exposure and their concentrations in biological matrices in rats. A previous study found that the concentrations of five *REEs* (La, Ce, Pr, Nd, and Gd) in rat fur were positively correlated with their administration dose [24], whereas no such correlation was found in whole blood, which is consistent with our findings. However, our study provides additional confirmation of the utility of up to 14 *REEs* as biomarkers. Additionally, our study adopted an efficient fur washing protocol to avoid potential contamination of ambient particulate matter [23], and thus was able to provide more reliable results. In the literature, blood REE concentrations have been used as biomarkers in epidemiological studies [11,25]. However, REE concentrations measured in blood samples to evaluate exposure should be interpreted more carefully in the future. The reliability of using hair REE concentrations as exposure biomarkers was also confirmed in one of the largest REE mine areas in the Inner Mongolia region in China [21,26]. Additionally, hair REE concentrations have been more widely used in environmental health studies [9,27,28]. Overall, we conclude that REE concentrations in hair, compared with blood, represent a more reliable sample type for evaluating their exposure levels.

The association between population exposure to *REEs* and hypertension risk were first proposed in an epidemiological study conducted among a housewife population in Shanxi Province, China [9]. Similarly, the concentration of *REEs* in hair has been positively correlated with indoor air pollution [8]. This study provided a clue to elucidate the potential effect of REE exposure on hypertension development. In this study, we provide a confirmation of the reliability of using hair samples to represent the internal intake of *REEs* within a population. However, this study does not provide evidence allowing conclusion regarding humans on the basis of animal models. In our follow-up study, we intend to demonstrate that it is reasonable to use hair samples to evaluate the long-term exposure of a population, as performed in the previous study [9]. However, these rat model results do not support the hypothesis that REE exposure can affect changes in blood pressure. In two recently published studies, it was suggested that La(NO_3_)_3_, one of the typical light *REEs* in nitrate form, is able to ameliorate atherosclerosis by regulating lipid metabolism, oxidative stress, endothelial dysfunction and inflammatory response in mice [16], and that it could even restore the redox homeostasis disrupted by ethanol by provoking Keap 1/Nrf2/p62 signaling pathway [17]. As atherosclerosis is an important cause of hypertension [19], these two studies also do not support the hypothesis that REE intake can affect blood pressure.

A number of REE elements play a role in the electron-exchange reactions in living organisms, causing the generation of free radicals that affect biological processes, e.g., oxidative stress [29]. In the previous study, it was demonstrated that *REEs* had a hormetic effect on oxidative stress. In an animal study, it was found that acute lung injury following intratracheal instillation of YCl_3_, a REE chloride, could increase the Mn-SOD level, enhancing the anti-oxidative capacity [30]. Furthermore, another study found that LaCl_3_ was able to protect the hepatic morphological alternations of ethanol model mice and reduce the malondialdehyde (MDA) level, suggesting that low doses of REE exert antioxidative effects [17]. However, high doses of La, Ce, and Nd can significantly decrease the superoxide dismutase (SOD), catalase (CAT), and glutathione peroxidase (GPx) activities and GSH levels of hepatocyte mitochondria, indicating their protective role in balancing damage resulting from oxidative stress [31]. In our study, we did not find an association between plasma 8-OHdG concentration and administration dose of REE mixture at relatively high exposure levels, indicating that *REEs* may not play a key role in oxidative stress response in rats. 8-OHdG is also an important oxidative stress biomarker. An animal study suggested that 8-OHdG was associated with the hypertension [32], and similar results were also observed in a human study [33]. As it is widely known from previous studies, oxidative stress damage is an important biological process that interferes with blood pressure [34] and impacts the development of hypertension [35], we speculated that *REEs* potentially do not significantly contribute to the development of hypertension through the oxidation pathway. Further in-depth studies should be conducted in order to confirm our findings.

### Limitations, Advantages, and Implications

Our study has certain limitations that need to be addressed. Overall, we only evaluated the changes of body weight, blood pressure, and oxidative stress after three-month exposure. The study results should be extrapolated with care. Furthermore, we used intragastric administration instead of inhalation to evaluate exposure to *REEs*, and this may not correspond to the actual exposure scenario. However, the current study exhibits several strengths in terms of elucidating the toxic effects of various *REEs*. First, our study confirmed the reliability of using hair *REEs* as exposure biomarkers. We showed that blood samples may not be an appropriate matrix for illustrating REE exposure levels in a population. Second, the current study is the first to explore the potential relationship between REE exposure and blood pressure change using a rat model. China has relatively abundant REE reserves and has been the dominant supplier in the world REE market in recent decades. It has been recognized that large amounts of *REEs* have been released into the surrounding ecosystems due to extensive mining and refining activities. Our study suggests that the effects of REE exposure on hypertension, as one of the most important diseases in China, may be weak. However, more in-depth study should be conducted to confirm our findings.

## 5. Conclusions

We conclude that hair samples constitute an efficient matrix for indicating population exposure to *REEs*, and *REEs* may have no direct effect on the blood pressure change using this rat model.

## Figures and Tables

**Figure 1 ijerph-18-09836-f001:**
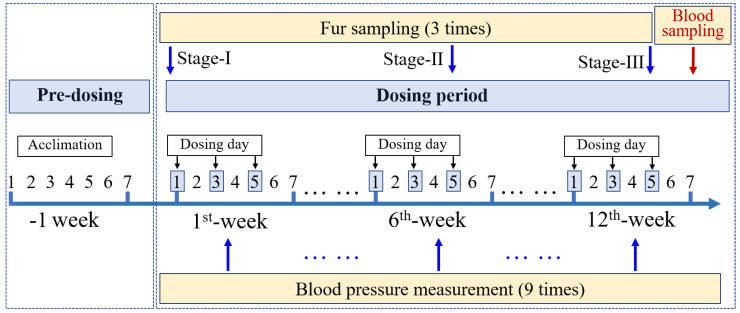
A schematic of the study design of the animal experiment.

**Figure 2 ijerph-18-09836-f002:**
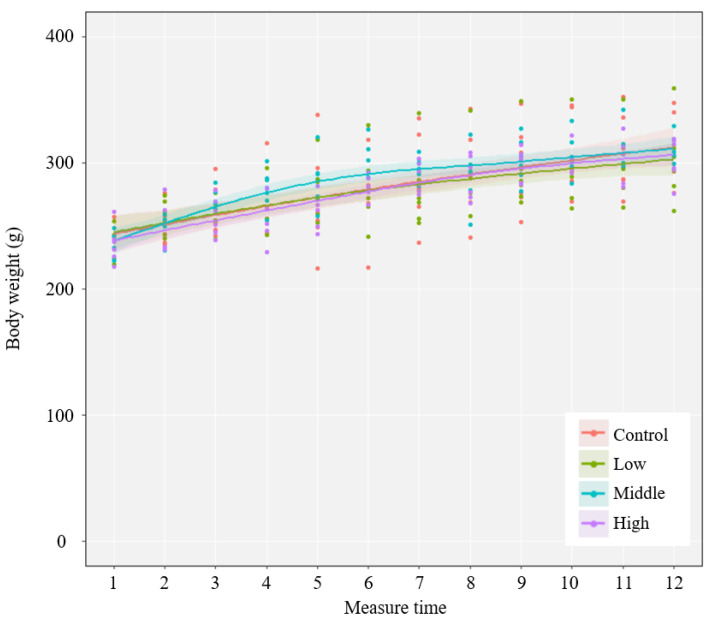
The effect of rare earth elements’ administration on body weight changes of rats among the four administration groups, i.e., control, low, middle, and high. The change curve was fitted using a non-linear regression model with R software. The shaded area represents the 95% confidence level.

**Figure 3 ijerph-18-09836-f003:**
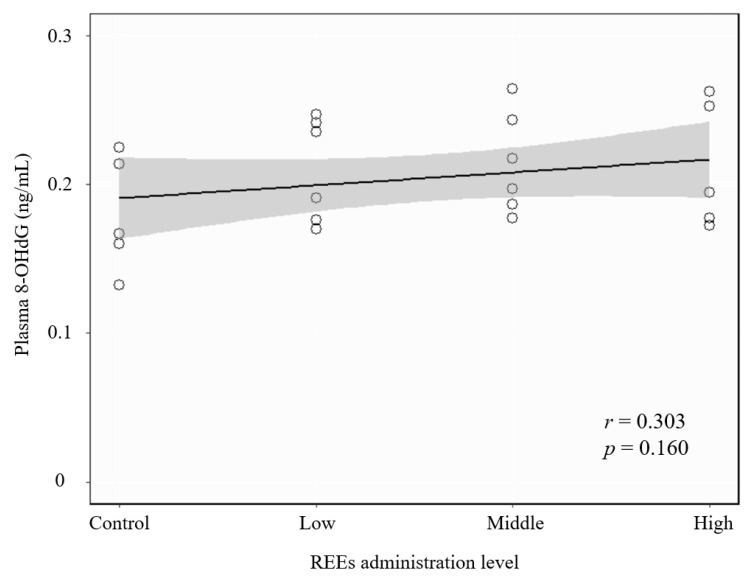
Spearman correlation between the administration dose level of *REEs* to rats and 8-OHdG concentration in rat plasma. The dose–response curve is fitted by using linear regression model with R software. The shaded area represents the 95% confidence level.

**Table 1 ijerph-18-09836-t001:** The dose–response relationships between administration levels of 14 *REEs* to rats and their concentrations in rat fur samples collected in three stages.

*REEs*	Control	Low	Middle	High	*P* _trend_ ^c^
Unit: ng/g rat’s fur	Stage-I				
La	1349 ± 1088 ^a^	505 ± 218	400 ± 164	497 ± 364	0.031
Ce	4308 ± 2460	1661 ± 397	975 ± 290	6095 ± 7335	0.972
Pr	364 ± 395	/	209 ± 207	117.18	/
Nd	1139 ± 1332	/	/	196 ± 155	/
Sm	208 ± 149	112 ± 45.8	213 ± 308	126 ± 43.9	0.776
Eu	746 ± 868	334 ± 253	706 ± 944	218 ± 127	0.318
Gd	545 ± 719	/	329.11	538	/
Tb	509 ± 527	97.9 ± 58.2	291 ± 241	193 ± 192	0.211
Dy	254 ± 153	66.4 ± 42.9	187 ± 224	56.7 ± 47.7	0.124
Ho	104 ± 34.0	49.7 ± 11.7	135 ± 213	34.6 ± 14.9	0.550
Er	114 ± 74.75	107 ± 140	156 ± 216	34.2 ± 28.6	0.446
Tm	24.548 ± 13.9	5.86	189 ± 277	22.8 ± 12.2	0.952
Yb	94.7 ± 74.4	26.1 ± 15.1	204 ± 272	40.4 ± 30.7	0.985
Y	1002 ± 486	423.5 ± 138.3	480 ± 227	285 ± 205	0.001
Σ*REEs* ^b^	9501 ± 6389	2887 ± 504	3482 ± 2173	4208 ± 6492	0.079
	Stage-II				
La	1068 ± 375	1266 ± 624	1725 ± 622	6368 ± 3063	0.002
Ce	2214 ± 1151	2281 ± 1138	2303 ± 558	5848 ± 3094	<0.001
Pr	295 ± 148	444 ± 447	326 ± 111	1353 ± 782	<0.001
Nd	792 ± 242	1504 ± 1575	1230 ± 500	5185 ± 3109	<0.001
Sm	179 ± 93.3	177 ± 71.1	269 ± 92	1033 ± 595	0.009
Eu	257 ± 138	124 ± 37.9	255 ± 179	433 ± 156	0.005
Gd	273 ± 180	457 ± 539	344 ± 130	1629 ± 928	0.006
Tb	185 ± 218	246 ± 69.3	167 ± 49.8	297 ± 96.5	0.034
Dy	166 ± 106	156 ± 68.9	249 ± 76	962 ± 439	0.004
Ho	29.5 ± 15.7	28.4 ± 10.5	43.6 ± 14.7	175 ± 61.3	0.002
Er	98.4 ± 48.5	108 ± 38.9	163.1032 ± 57.7	683 ± 305	0.029
Tm	16.5 ± 12.0	16.3 ± 8.88	21.5 ± 6.06	107 ± 37.9	<0.001
Yb	123 ± 107	113 ± 47	168 ± 60.3	686 ± 303	0.003
Y	549 ± 113	747 ± 236	924 ± 314	2541 ± 1049	0.018
Σ*REEs*	5938 ± 2129	7513 ± 3952	8252 ± 2039	30,693 ± 149,23	0.007
	Stage-III				
La	497 ± 205	904 ± 942	417 ± 229	3927 ± 1476	0.002
Ce	542 ± 337	1398 ± 1349	948 ± 424	3717 ± 1358	<0.001
Pr	115 ± 83	195 ± 220	407 ± 215	1255 ± 407	<0.001
Nd	518 ± 324	918 ± 1100	1701 ± 905	5098 ± 1397	<0.001
Sm	55.4 ± 47.4	347 ± 309	32.9 ± 32.7	693 ± 330	0.009
Eu	/	129	131 ± 137	316 ± 96	/
Gd	253 ± 123	243 ± 250	579 ± 377	1636 ± 602	<0.001
Tb	1338 ± 2216	110 ± 67.9	733 ± 759	651 ± 621	0.464
Dy	150 ± 151	66.8 ± 65.8	60.1 ± 34.9	638 ± 286	0.004
Ho	44.9	1.65	/	76.2 ± 37.3	/
Er	65.8 ± 87.4	36.5	16.5 ± 21.6	338 ± 158	0.029
Tm	3.5	/	/	46.6 ± 28.6	/
Yb	70.1 ± 84.4	14.3 ± 11.2	30.9 ± 21.7	354 ± 131	0.003
Y	500 ± 697	218 ± 175	230 ± 108	1326 ± 443	0.018
Σ*REEs*	3285 ± 2745	5245 ± 7424	4891 ± 1952	18,732 ± 6630	0.001

^a^ Data are shown as arithmetical mean or arithmetical mean ± standard deviation, unit: ng/g rat fur; ^b^ Sum of REE concentration; ^c^ Statistical test by linear regression analysis, and values < 0.05 are marked in red.

**Table 2 ijerph-18-09836-t002:** The dose–response relationships between administration levels of 14 *REEs* to rats and their concentrations in rat plasma and blood cell samples.

*REEs*	Control	Low	Middle	High	*P* _trend_ ^c^
	Plasma (unit: ng/mL)				
La	69.8 ± 9.02 ^a^	70.2 ± 48.4	89.7 ± 90.6	78.9 ± 56.1	0.655
Ce	209 ± 131	261 ± 265	669 ± 446	205 ± 165	0.497
Pr	14.5 ± 9.12	20.8 ± 20.9	67.0 ± 39.2	26.6 ± 16.2	0.168
Nd	82.7 ± 82.1	54.1 ± 64.4	287 ± 291	78.1 ± 49.3	0.663
Sm	14.0 ± 4.60	12.7 ± 10.4	23.6 ± 19.3	15.6 ± 12.4	0.509
Eu	3.29 ± 3.68	3.80 ± 4.28	10.9 ± 7.60	5.38 ± 4.83	0.414
Gd	23.1 ± 18.2	14.1 ± 19.5	40.4 ± 33.8	13.5 ± 7.72	0.853
Tb	26.5 ± 13.5	23.3 ± 19.5	73.9 ± 131	110 ± 110	0.128
Dy	10.3 ± 4.88	11.0 ± 9.40	24.1 ± 23.5	16.3 ± 7.40	0.235
Ho	6.43 ± 3.34	11.7 ± 3.10	19.4 ± 5.22	21.9 ± 3.48	<0.001
Er	5.03 ± 4.08	9.29 ± 10.8	19.1 ± 20.0	9.44 ± 7.65	0.473
Tm	10.3 ± 10.3	3.60 ± 0.570	4.11 ± 2.01	1.39 ± 0.90	0.032
Yb	4.11 ± 4.30	46.8 ± 57.3	14.1 ± 18.9	13.7 ± 9.97	0.857
Y	85.1 ± 70.5	58.7 ± 40.3	82.3 ± 74.6	49.8 ± 28.2	0.420
*ΣREEs* ^b^	526 ± 266	468 ± 413	1167 ± 833	583 ± 231	0.394
	Blood cell (unit: ng/g)				
La	338 ± 307	222 ± 129	314 ± 199	284 ± 193	0.655
Ce	2287 ± 1659	963 ± 596	1482 ± 730	1268 ± 676	0.497
Pr	116 ± 108	87.8 ± 53.8	97.0 ± 61.5	79.5 ± 59.9	0.168
Nd	334 ± 334	334 ± 189	319 ± 148	255 ± 162	0.663
Sm	81.8 ± 74.8	57.2 ± 35.3	66.3 ± 45.1	51.0 ± 34.3	0.352
Eu	45.7 ± 35.9	24.6 ± 15.5	31.6 ± 17.8	17.6 ± 10.6	0.235
Gd	176 ± 170	106 ± 65.2	215 ± 115	113 ± 70.7	0.685
Tb	477 ± 224	296 ± 178	1095 ± 412	658 ± 500	0.347
Dy	57.9 ± 53.4	44.1 ± 29.6	52.1 ± 27.8	34.1 ± 19.5	0.313
Ho	9.17 ± 9.57	5.00 ± 3.17	5.85 ± 4.62	3.72 ± 1.93	0.179
Er	30.5 ± 25.0	23.3 ± 13.7	25.9 ± 15.1	20.5 ± 10.6	0.365
Tm	10.9 ± 7.48	6.39 ± 3.94	14.0 ± 9.75	4.94 ± 2.92	0.428
Yb	27.3 ± 27.7	19.1 ± 15.2	29.9 ± 18.7	17.9 ± 6.64	0.650
Y	306 ± 143	365 ± 251	536 ± 481	213 ± 91.5	0.420
*ΣREEs*	4297 ± 2740	2129 ± 1240	4101 ± 1168	3015 ± 1501	0.598

^a^ Data are shown arithmetical mean ± standard deviation, unit: ng/g rat fur; ^b^ Sum concentration of *REEs*; ^c^ Statistical test by linear regression analysis, and values < 0.05 are marked in red.

**Table 3 ijerph-18-09836-t003:** Dose–response analysis between the administration levels of 14 *REEs* to rats and blood pressure.

Groups	*β*	*p*	*β*	*p*
	Systolic blood pressure		Diastolic blood pressure	
Control	0		0	
Low	9.93 (−1.54–21.4)	0.090	10.2 (−8.30–28.7)	0.279
Middle	−9.11 (−20.6–2.36)	0.119	−7.41 (−25.9–11.1)	0.433
High	5.26 (−6.21–16.7)	0.369	8.67 (−9.85–27.2)	0.359
	Heart rate		Pulse pressure	
Control	0		0	
Low	13.6 (−25.3–52.4)	0.494	−0.11 (−9.24–9.02)	0.981
Middle	−23.2 (−62.0–15.6)	0.242	−1.44 (−10.6–7.68)	0.756
High	14.8 (−24.1–53.6)	0.456	−3.33 (−12.5–5.80)	0.474

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
