# Peer review of "Effects of Rare Earth Elements on Blood Pressure and Their Exposure Biomarkers: Evidence from Animal Experiments"

_ijerph, 2021, doi:10.3390/ijerph18189836_

Round 1
Reviewer 1 Report
Dear Author,
Below please find some suggestions and questions:
- It is clear that fur samples seem to be a better biomarker of REEs exposure than blood or plasma samples. For the fur samples collected at both Stage-II and Stage-III, there were overall significant dose–response relationships of concentrations for the 14 administered REEs. On the other hand, no dose–response relationships were observed between the doses of the 14 REEs administered to rats and their concentrations in either the plasma or blood samples.
There is just one unusual finding. The total concentration of REEs in the control group is much higher than total concentration in the exposure groups. Could you please try to explain this phenomenon?
- There was not observed a dose-response relationship between the rats’ REE administration dose and blood pressure neither the correlation between REE administration dose and 8-OHdG concentration in rat plasma. Both finding are valid for the intragastric dosing.
Fly ash from coal combustion contains a high amount of REEs and it is a significant source of air pollution in northern China. In the article the administration route was intragastric. Based on the main source of pollution, ash, the inhalation route shall be considered as a more appropriate way for REEs administration.
Author Response
1、It is clear that fur samples seem to be a better biomarker of REEs exposure than blood or plasma samples. For the fur samples collected at both Stage-II and Stage-III, there were overall significant dose–response relationships of concentrations for the 14 administered REEs. On the other hand, no dose–response relationships were observed between the doses of the 14 REEs administered to rats and their concentrations in either the plasma or blood samples.
There is just one unusual finding. The total concentration of REEs in the control group is much higher than total concentration in the exposure groups. Could you please try to explain this phenomenon?
Response: Thank you for your concern. For such results, we have double-checked the raw data and there is nothing wrong about the measurement. We guess it should be an outlier. However, we cannot explain it using the current information and still keep it in the manuscript. It will not affect the overall conclusion.
2、There was not observed a dose-response relationship between the rats’ REE administration dose and blood pressure neither the correlation between REE administration dose and 8-OHdG concentration in rat plasma. Both finding are valid for the intragastric dosing.
Fly ash from coal combustion contains a high amount of REEs and it is a significant source of air pollution in northern China. In the article the administration route was intragastric. Based on the main source of pollution, ash, the inhalation route shall be considered as a more appropriate way for REEs administration.
Response: Thanks for the question, Considering the actual situation of air pollution in the northern China, inhalation is exactly the best exposure pattern. But inhalation can not precisely control the exposure level, therefore, we choose the intragastric dosing. We will add to this discussion in the part of limitation.
Reviewer 2 Report
This paper (ijerph-1354386) conducted rats experiment models of REEs exposure and assessed more reliable sample matrix of REEs exposure, the association between REEs exposure and hypertension, and oxidative stress. It figured out the fur / hair is more suitable for monitoring REEs exposure rather than blood sample; while some statements should be more accurate and structure need to be reorganized, and the logical expression should be improved. I suggest reconsidering its publication after its major revisions.
- This paper tried to evaluate three different things (sample matrix, hypertension, oxidative stress) about REEs exposure. But the authors cannot make them correlated with each other, which made the paper confusing to readers.
- L33 Abstract: “the exposure biomarker” should be revised to be “an exposure biomarker”.
- Abstract: Please specify how many REEs were administered. Too many words about methods in abstract. Please remove or modify the methods parts, and add more details about results.
- L60: “there”?
- L71: The authors have discussed multiple mechanisms between hypertension and REE. While this sentence still expressed out there’s a knowledge gap. So please modify the logic and expression.
- L77: The aims are to figure out biomarker of REEs. But the sentences before the aims were talking about biological matrix. Please make the links. I think the authors should revise this paragraph even the whole “Introduction” section.
- L97: the authors missed neodymium (Nd) here.
- L99-100: please specify the doses and concentrations.
- Figure 1: “6h week”?
- L135: “The Ge concentration”? So REEs in fur detected was “Ge” or the same as blood sample?
- L138-142: Please describe more detailed information about 8-OHdG measurement.
- Figure 2: please specify “blank, low, middle, high”.
- L167-168, L179: what is the unit? ng/g or ng/mg? It’s better to list the unit just in table, not in the note under table.
- L176: The authors should make clear the definition of biomarkers. All the results are actually talking about reliable sample type, not biomarkers.
- L184: unit: ng/g rat’s fur?
- Table 1 and 2: please keep the titles of table 1 and 2 uniform. Please specify the statistical analysis method and list it in section 2.4 data analysis.
- L194-198: Have the authors considered some other confounders for the association between REEs administration and 8-OHdG?
- L209: 14 REEs?
- L207-221: Please make clear the definition of biomarker and modify it through the paper.
- Table 3: why is the blood pressure of control 0? What’s the unit? What does the “p” mean? If the authors compared each administration group to control group, please don’t use linear mixed-effects model.
- L232-237: the references interpretation and the conclusion of “these two studies also did not support that REE intake can affect the blood pressure” is far-fetched. The authors should find more related articles or explain more about these two references.
- Please improve the language and revise errors through the paper.
Author Response
1、This paper (ijerph-1354386) conducted rats experiment models of REEs exposure and assessed more reliable sample matrix of REEs exposure, the association between REEs exposure and hypertension, and oxidative stress. It figured out the fur / hair is more suitable for monitoring REEs exposure rather than blood sample; while some statements should be more accurate and structure need to be reorganized, and the logical expression should be improved. I suggest reconsidering its publication after its major revisions.
Response: Thanks very much for your nice consideration. The one-to-one response has been provided as bellows.
2、- This paper tried to evaluate three different things (sample matrix, hypertension, oxidative stress) about REEs exposure. But the authors cannot make them correlated with each other, which made the paper confusing to readers.
Response: It’s important to evaluate the exposure of REEs, therefore, our study the discuss the performance of fur and blood sample matrix, for further screening for reliable biological markers.
In our study, another main research objective is hypertension, and oxidative stress is just an important pathway to explain the reason why the hypertension happens.
3、- L33 Abstract: “the exposure biomarker” should be revised to be “an exposure biomarker”.
Response: Revised accordingly. Please see the lines 38 In the revised manuscript.
4、- Abstract: Please specify how many REEs were administered. Too many words about methods in abstract. Please remove or modify the methods parts, and add more details about results.
Response: A total of 14 REEs was administered, and we add the information in the abstract. See line 22. Some part of method has been removed, and we add more details about result. We further add it in the abstract as “Hair samples are an efficient matrix for indicating population exposure REEs.”and” REEs may have no effect on the blood pressure change.”. Please see line 32 and 34.
5、- L60: “there”?
Response: Revised accordingly. Please see the lines 68 in the revised manuscript.
6、- L71: The authors have discussed multiple mechanisms between hypertension and REE. While this sentence still expressed out there’s a knowledge gap. So please modify the logic and expression.
Response: Because of few animal experiment evidence, we can just only discuss the possible mechanisms and there is no conclusive evidence to demonstrate the relationship between REEs and hypertension.
7、- L77: The aims are to figure out biomarker of REEs. But the sentences before the aims were talking about biological matrix. Please make the links. I think the authors should revise this paragraph even the whole “Introduction” section.
Response: We revise the part of introduction and add below sentence “In vitro experiment indirectly proved the possible association of REEs and cardiovascular disease, but few epidemiology study and no animal model evidence demonstrated it.” in this section. Please see in the revised manuscript.
8、- L97: the authors missed neodymium (Nd) here.
Response: Revised accordingly.
9、- L99-100: please specify the doses and concentrations.
Response: The dose was shown in the Table S1, and we will also revise the sentence to add the information of dose and concentrations.
10、- Figure 1: “6h week”?
Response: Should be 6th, thank you for the reminder.
11、- L135: “The Ge concentration”? So REEs in fur detected was “Ge” or the same as blood sample?
Response: Thanks for your reminder. The mistake was corrected.
12、- L138-142: Please describe more detailed information about 8-OHdG measurement.
Response: The detailed measurement procedure was added. We further add the sentence ”The measure theory is ELISA method based on the competitive inhibition immunochromatography mechanism.”See line 160-161
13、- Figure 2: please specify “blank, low, middle, high”.
Response: The detailed explanation was added in the footnote. See line 185
14、- L167-168, L179: what is the unit? ng/g or ng/mg? It’s better to list the unit just in table, not in the note under table.
Response: Thank you for the suggestion. The unit is ng/g, and we revised accordingly.
15、- L176: The authors should make clear the definition of biomarkers. All the results are actually talking about reliable sample type, not biomarkers.
Response: Thank you for the suggestion, we modify this definition and see in the revised manuscript. Biomarkers are the concentration of a target substance in an exact sample matrix, and when we talk about biomarkers, the sample matrix cannot neglect.
16、- L184: unit: ng/g rat’s fur?
Response: Yes, the unit is ng/g rat’s fur.
17、- Table 1 and 2: please keep the titles of table 1 and 2 uniform. Please specify the statistical analysis method and list it in section 2.4 data analysis.
Response: Thank you for the suggestion. We refine the section of data analysis. We further add the “Linear regression was used to analyze the concentration trend of different exposure level.” in this part.
18、- L194-198: Have the authors considered some other confounders for the association between REEs administration and 8-OHdG?
Response: Thank you for the suggestion. In this study, we did not consider the confounder, because the there is no confounder we can choose to adjust the association.
19、- L209: 14 REEs?
Response: Revised accordingly.
20、- L207-221: Please make clear the definition of biomarker and modify it through the paper.
Response: Thank you for the suggestion, and the definition of biomarker was re-corrected.
21、- Table 3: why is the blood pressure of control 0? What’s the unit? What does the “p” mean? If the authors compared each administration group to control group, please don’t use linear mixed-effects model.
Response: control 0 is the reference, and the 0 means the β is 0 and showed no influence in the formula. P is the probability calculated by linear mixed-effect regression analysis. In this repeated measure design experiment, linear mixed-effects model is a good way to explore the association between exposure and outcomes.
22、- L232-237: the references interpretation and the conclusion of “these two studies also did not support that REE intake can affect the blood pressure” is far-fetched. The authors should find more related articles or explain more about these two references.
Response: We agree with you, but these are only two relevant studies we can find, and more related articles have not been reported yet.
23、- Please improve the language and revise errors through the paper.
Response: Thank you for the reminder and we polish the article and see the new manuscript.
Reviewer 3 Report
The authors report on their valuable study of REE exposure in rats, which may raise the interest of Journal's readers and may be possibly eligible for acceptance.
Unfortunately, the manuscript is littered of errors in English style and grammar, thus it requires a proofreading work from a better trained English expert.
It also requires some important corrections on the expression of the data.
Please correct as indicated in the following.
Line 82, line 96, line 98: chlorinated REEs.
I never saw the use of the term chlorinated for inorganic compounds; please correct with “REEs chlorides” or an equivalent term.
Line 96-100
The concentrations of the prepared solutions are not indicated. Please indicate the concentration ranges of the dose to be administered for the low, medium and high range.
From table S1 it can be seen that the concentrations are different for the different REEs, with what criteria were the different concentrations chosen?
Line 124-136
The values of LOD and LOQ were not calculated for each element in the two matrices (blood and fur). Recovery, precision and accuracy data are not reported.
Table S3 shows the comparison data with the certified material, it would be advisable to process the data obtained. For the blood matrix the use of certified reference materials is not observed; it would recommended to add it.
Cerium (Ce) 20.93 ± 1.15: too many significant figures: please use the correct number here and in the other tables.
Table 1
The reported values are average values; over how many replicates were they calculated?
Table 1 shows anomalous values of mean value and standard deviation. For example: cerium, stage I at high concentration 6095 ± 7335. The standard deviation is very high. Many of these data are seen in Table 1. Please check the data. Also check the data in table 2.
When you have high standard deviation values, it means that the replication values are very different. So it could mean that the analysis method is not reproducible. It is recommended to improve the quantitative analysis.
Author Response
1、The authors report on their valuable study of REE exposure in rats, which may raise the interest of Journal's readers and may be possibly eligible for acceptance.
Response: Thanks very much for your nice consideration. The one-to-one response has been provided as bellows.
2、Unfortunately, the manuscript is littered of errors in English style and grammar, thus it requires a proofreading work from a better trained English expert.
Response: Thank you for the reminder and we polish the article and see the revised manuscript.
3、It also requires some important corrections on the expression of the data.
Response: We agree with you. The term chlorinated REEs are corrected, and we will use REEs chlorides to replace them.
5、Line 96-100
The concentrations of the prepared solutions are not indicated. Please indicate the concentration ranges of the dose to be administered for the low, medium and high range.
From table S1 it can be seen that the concentrations are different for the different REEs, with what criteria were the different concentrations chosen?
Response: Thank you for this question. The criteria of the concentration were based on the paper, Dietary intake and burden of lanthanide in main organs and tissues for Chinese man by Hongda Zhu.In this paper, the author provided the human estimated daily intake(EDI) of REEs, and based on the data, we calculated the different concentration, low: 200 times of EDI; middle:2000 times of EDI and high:20000 times of EDI. These are very high dose. Each intragastric dose volume is 2 mL, and we can calculate the concentration range of different level.
6、Line 124-136
The values of LOD and LOQ were not calculated for each element in the two matrices (blood and fur). Recovery, precision and accuracy data are not reported.
Response: Thank you for the reminder. The information of LOD and LOQ were added in supplementary material.
7、Table S3 shows the comparison data with the certified material, it would be advisable to process the data obtained. For the blood matrix the use of certified reference materials is not observed; it would recommended to add it.
Response: There is no certified reference blood matrix materials for us to detect it. therefore, we cannot provide this information. In our study, all experiment data was obtained from the China metrology accreditation laboratory in CAS, and have several quality control method including parallel sample detection, etc. All the data are accurate and robust.
8、Cerium (Ce) 20.93 ± 1.15: too many significant figures: please use the correct number here and in the other tables.
Response: Thank you for the reminder, we revised accordingly.
9、Table 1
The reported values are average values; over how many replicates were they calculated?
Response: The average value was calculated by arithmetic mean value, and none replicate was calculated.
10、Table 1 shows anomalous values of mean value and standard deviation. For example: cerium, stage I at high concentration 6095 ± 7335. The standard deviation is very high. Many of these data are seen in Table 1. Please check the data. Also check the data in table 2.
Response: We recheck the data and do not find any mistakes. The SD is high due to the animal experiment per se. Thanks for your understanding.
11、When you have high standard deviation values, it means that the replication values are very different. So it could mean that the analysis method is not reproducible. It is recommended to improve the quantitative analysis.
Response: Thank you for your suggestion. For each sample, the analytical results are robust. We consider that the high SD due to the variance of the rats. In the future, more replicates of the rats should be used to reduce the SD. The limitation has been added in the revised manuscript, “In our study, there are large variance among the rats in one group. We recommend that more replicates of the rats should be adopted in one group.”
Round 2
Reviewer 2 Report
The authors have provided responses and revised the manuscript to address the comments. While, the manuscript should be revised more carefully. Please see the comments listed below.
- L65: “thses”? The authors indicated the wrong line number, and revised to another wrong word, which is not expected!
- L78-79: please delete “Therefore, there is a knowledge gap in the relationship between the REE exposure and hypertension development.”
- L107: “. which exposure saline solution (control), 200 times of human estimated daily intake of REEs(low),2000 times(middle) and 20000 times(high).” ? Please make this kind of error as less as possible and read carefully through the paper.
- L24, 107-108, 176: Please add space between “REEs(low)”, “times(middle)” and “times(high).”
- L178: Please indicate the higher concentration is Y’s.
- L180-181: the authors revised the data in table 1. Why are the descriptions still same? Please modify this sentence “and even higher than the controls groups at Stage-II and Stage-III.”
- L182-183: Since the authors revised the data in rat’s fur. I didn’t see any significant dose-response relationships among the four groups except stage III. But the authors can try the linear regression between the exposure time and the concentrations in each group.
- L190: ng/g rat’s fur)? Same errors in the other tables.
- Please polish the manuscript again and more carefully, if the authors have already done that.
Author Response
The authors have provided responses and revised the manuscript to address the comments. While, the manuscript should be revised more carefully. Please see the comments listed below.
Response:Thanks very much for your nice consideration. The one-to-one response has been provided as bellows.
1、- L65: “thses”? The authors indicated the wrong line number, and revised to another wrong word, which is not expected!
Response:Sorry for this mistake, we have revised accordingly. Please see line 65.
2、- L78-79: please delete “Therefore, there is a knowledge gap in the relationship between the REE exposure and hypertension development.”
Response: Thank you for the suggestion and revised accordingly.
3、- L107: “. which exposure saline solution (control), 200 times of human estimated daily intake of REEs(low),2000 times(middle) and 20000 times(high).” ? Please make this kind of error as less as possible and read carefully through the paper.
Response: Thanks for your reminder and we revised.
Please see line 110~111
4、- L24, 107-108, 176: Please add space between “REEs(low)”, “times(middle)” and “times(high).”
Response: Revised accordingly.
-5、 L178: Please indicate the higher concentration is Y’s.
Response: Thank you for the suggestion, we indicate the concentration of Y element, please see line 180~181.
6、- L180-181: the authors revised the data in table 1. Why are the descriptions still same? Please modify this sentence “and even higher than the controls groups at Stage-II and Stage-III.”
Response: Thank you for the suggestion. After we rechecked the data and revised the data in Table 1, the data conclusion remained the same, therefore we did not modify the manuscript. After the revision, the Y concentration of control group in stage I still higher than that in stage II and stage III.
7、- L182-183: Since the authors revised the data in rat’s fur. I didn’t see any significant dose-response relationships among the four groups except stage III. But the authors can try the linear regression between the exposure time and the concentrations in each group.
Response: Thank you for the suggestion. In Table 1, there was dose-response relationship for all the REE by using a linear regression analysis in Stage-III. We re-analyze the data and correct the results. In this study, we aimed to explore the relationship between the administration level of REE and their hair concentrations in Stage-II and Stage-III. Overall, there were certain increase of hair REE concentrations in both Stage-II and Stage-III compared with those in Stage-I. As the hair concentrations of REEs mainly depend on the exposure level, we considered that it is necessary to analyze the relationship between hair REEs and exposure period.
-8、 L190: ng/g rat’s fur)? Same errors in the other tables.
Response: Revised accordingly.
-9、 Please polish the manuscript again and more carefully, if the authors have already done that.
Response: Thank you for the reminder, we re-polished the manuscript by another expert.